# Application of Host-Depleted Nanopore Metagenomic Sequencing in the Clinical Detection of Pathogens in Pigs and Cats

**DOI:** 10.3390/ani13243838

**Published:** 2023-12-13

**Authors:** Xu Han, Zhaofei Xia

**Affiliations:** College of Veterinary Medicine, China Agricultural University, Beijing 100193, China

**Keywords:** nanopore sequencing, metagenomic sequencing, host gene removal, veterinary pathogenic microorganisms

## Abstract

**Simple Summary:**

Metagenomic sequencing, as an emerging detection technology, has the capability to non-specifically detect various bacteria and viruses. However, a series of issues have limited its application in animal clinical diagnostics. In response to these challenges, this study focused on clinical samples from cats and pigs and optimized sequencing and analysis methods. The results demonstrated that the optimized nanopore metagenomic sequencing method could effectively detect bacteria and viruses in the samples while reducing interference from host genes. When combined with real-time cloud-based and local analysis, it enabled the quick determination of when sequencing should be terminated and facilitated the assembly of relatively complete viral genomes. This method is characterized by its low equipment requirements, short sequencing time, and cost-effectiveness, making it suitable for deployment in small- to medium-sized animal clinical testing laboratories.

**Abstract:**

Metagenomic sequencing is a valuable tool for non-specifically detecting various microorganisms in samples, offering unique advantages for detecting emerging pathogens, fastidious or uncultivable pathogens, and mixed infections. It has recently been applied to clinically detect pathogenic microorganisms in animals; however, the high proportion of host genes, expensive sequencing equipment, and the complexity of sequencing and data analysis methods have limited its clinical utility. In this study, a combination of tissue homogenization and nuclease digestion was employed to remove host genes from pig and cat samples; DNA and RNA were then extracted and subjected to nonselective PCR amplification to simultaneously detect DNA and RNA pathogen genomes using R9.4.1 or R10.4.1 flow cells on the MinION platform. Real-time pathogen detection was conducted using EPI2M WIMP, and viral genome assembly was performed using NanoFilt, minimap2, samtools, and ivar. Pathogens in five clinical samples (serum, nasopharyngeal swab, feces, or ascites) from cats and four clinical samples (lung or small intestine tissue) from pigs were examined by metagenomic sequencing, and the results were consistent with those obtained by PCR and bacterial culture. Additionally, we detected four viruses and three bacteria that may be associated with diseases. A comparison of results before and after host gene removal in three samples showed a 9–50% reduction in host genes. We also compared the assembly efficiency of six virus genomes and found that data volumes ranging from 3.3 to 98.3 MB were sufficient to assemble >90% of the viral genomes. In summary, this study utilized optimized nanopore metagenomic sequencing and analysis methods to reduce host genes, decrease the required data volume for sequencing analysis, and enable real-time detection to determine when to stop sequencing. The streamlined sequencing and analysis process overcomes barriers to the veterinary clinical application of metagenomic sequencing and provides a reference for clinical implementation.

## 1. Introduction

Metagenomic sequencing is an indiscriminate method for sequencing all microorganisms present in a sample and can detect numerous infectious pathogens, including bacteria, fungi, and viruses [1,2,3]. It offers unique advantages for identifying uncommon pathogens and emerging infectious agents. For instance, the novel coronavirus (severe acute respiratory syndrome coronavirus 2) responsible for the 2019 outbreak was initially detected through metagenomic sequencing [4,5]. Similarly, metagenomic sequencing was instrumental in identifying porcine pseudorabies virus as the cause of previously unexplained cases of acute encephalitis in humans [6]. Retrospective studies on lower respiratory samples from children revealed that approximately half of the samples previously diagnosed as pathogen-negative through clinical diagnostics harbored potential pathogens [3].

Metagenomic sequencing has recently been applied in animal clinical diagnostics [7,8,9]. However, it presents challenges, such as expensive sequencing equipment, increased sequencing time and costs due to abundant host DNA, and complexity in sequencing preparation and data analysis [10,11].

One common issue in the metagenomic sequencing of clinical samples is that host genes comprise most of the data (e.g., up to 99.999% of the genomic information in sputum samples) [12,13], while the proportion of pathogens is relatively small; in some cases, only one or two reads originate from the pathogen [14]. Therefore, most sequencing data are uninformative. Various methods have been developed to reduce host genes and increase the proportion of pathogen genes in samples. Common host gene removal methods include mechanical grinding combined with nucleases [15,16], saponin and nuclease treatment [12,17,18,19], propidium monoazide (PMA) treatment [20,21], and commercial reagent kits [17,22]. Grinding combined with nucleases is typically used for viral detection, while saponin and commercial kits are often employed for bacterial detection. Clinical diagnostics require a universal method to simultaneously detect potential bacteria and viruses in samples.

Conventional next-generation sequencing platforms (e.g., Illumina and Ion Torrent) are expensive and often unaffordable for animal clinical diagnostic laboratories. Moreover, sending samples to sequencing companies may not meet the clinical need for timely results. The MinION sequencing platform is a third-generation nanopore sequencing device known for its affordability, making it suitable for deployment in animal clinical diagnostic laboratories.

Traditional sequencing methods involve sequencing and subsequent analysis. In contrast, nanopore sequencing platforms offer real-time sequencing and analysis capabilities, allowing users to monitor the detection of pathogens and their corresponding reads in real-time. Deciding when to stop sequencing based on real-time detection results is a valuable research question. The goal is to generate the minimum threshold of data to meet the subsequent analysis needs, preventing the requirement for resequencing due to insufficient data while minimizing sequencing chip consumption to save sequencing time and costs.

In this study, we employed optimized nanopore metagenomic sequencing and analysis methods to test various sample types from pigs and cats. The results revealed that this method could reduce host genes and detect various bacteria, DNA viruses, and RNA viruses in the samples. Additionally, we demonstrated the ability to determine the necessary sequencing time using real-time results. This research provides valuable insights for the future application of metagenomic sequencing in animal clinical diagnostics.

## 2. Materials and Methods

### 2.1. Sample Information

Clinical samples (serum, nasopharyngeal swab, feces, or ascites) from five cats were obtained from the Small Animal Hospital of China Agricultural University. All cats were referred for clinical examination due to suspected pathogenic infection. Clinical samples (lung or small intestine tissue) were also collected postmortem from pigs at four swine farms. Standard laboratory tests were conducted in addition to metagenomic sequencing, and the results were compared for consistency. Detailed information is presented in Table 1.

### 2.2. Metagenomic Sequencing

An overview of the metagenomic sequencing process is illustrated in Figure 1.

#### 2.2.1. Nuclease Digestion and Nucleic Acid Extraction

The nucleic acid extraction process was initiated by taking 1 mL of fresh liquid samples (e.g., serum, swab eluate, or ascites) or 1 mL of tissue freeze—thawed exudate (if the exudate was insufficient, 1X PBS was added, ensuring thorough contact with the tissue, and then 1 mL of the mixture was aspirated). Subsequently, the sample was centrifuged at 4300× *g* for 3 min, and 500 μL of the supernatant was collected into a 2 mL centrifuge tube. Host gene removal was performed by tissue homogenization and nuclease digestion following the protocol of Oxford Nanopore Technologies with appropriate modifications [15,16,23]. This approach involved introducing 500 μL 1.4 mm zirconium oxide grinding beads to the 2 mL centrifuge tube and performing bead-beating at 50 Hz for 3 min at 4 °C. Freeze–thawing and tissue homogenization were used to disrupt host cells. After grinding, the liquid sample or freeze—thawed tissue fluid was centrifuged at 4300× *g* for 3 min, and 210 μL supernatant was added to 10 μL HL-SAN (ArcticZymes Technologies, Troms, Norway) for digestion. The samples were incubated at 37 °C in a Dry Bath incubator for 30 min, followed by centrifugation at 4300× *g* for 3 min. The HL-SAN enzyme was utilized to digest free DNA and RNA. Then, 200 μL of the enzymatically digested liquid was used for RNA and DNA extraction using the PureLink Viral RNA/DNA Mini Kit (Thermo Fisher Scientific, Waltham, MA, USA). In this step, 5.6 μL carrier RNA was replaced with 5.6 μL linear acrylamide (5 mg/mL, Thermo Fisher Scientific), and the nucleic acids were eluted in 12 μL H_2_O. Post-digestion, the nucleic acid concentration in the samples is typically exceedingly low. The PureLink Viral RNA/DNA Mini Kit maximizes the concentration of nucleic acids in the extracted material, which can reduce the final extracted volume to as low as 10 μL. Additionally, carrier RNA from the original kit was replaced with linear acrylamide to prevent interference with subsequent steps.

RNA reverse transcription and PCR random amplification were performed using the Sequence-Independent, Single-Primer Amplification (SISPA) method with 5 μL of the eluted nucleic acids following the method described by Lewandowski et al. [24] with appropriate modifications. Finally, DNA PCR random amplification was conducted using 5 μL of the eluted sample.

#### 2.2.2. RNA Reverse Transcription and Second-Strand Synthesis

A mixture was prepared by combining 1 μL 60 μM SISPA-N9 primer (SISPA-N9: GTTTCCCACTGGAGGATANNNNNNNNN), 5 μL nucleic acid, and 0.5 μL 10 mM dNTP for a total volume of 6.5 μL. The mixture was incubated at 65 °C for 5 min and cooled at 4 °C for 2 min. Subsequently, 2 μL 5X SSIV Buffer, 0.5 μL 0.1 M DTT, 0.5 μL SuperScript IV Reverse Transcriptase (Thermo Fisher Scientific), and 0.5 μL H_2_O were added for a total volume of 10 μL. The mixture was incubated at 23 °C for 10 min, 50 °C for 10 min, and 4 °C for 2 min. Then, 1 μL RNase H (Thermo Fisher Scientific) was added, and the sample was incubated at 37 °C for 20 min. Lastly, 1.5 μL NEB buffer 2, 1.5 μL H_2_O, and 1 μL Klenow Fragment (3′→5′ exo-) (New England Biolabs, Ipswich, MA, USA) were added for a total volume of 15 μL, and the sample was incubated at 37 °C for 40 min.

#### 2.2.3. Addition of Random Primers to Both Ends of DNA

After HL-SAN enzyme digestion, DNA concentrations were typically too low for direct sequencing library preparation. PCR random amplification was necessary, and random primers were added to both ends of the DNA strands before amplification.

A mixture was prepared by combining 5 μL nucleic acid, 1 μL 60 μM SISPA-N9 primer, and 0.5 μL 10 mM dNTPs for a total volume of 6.5 μL. The mixture was incubated at 95 °C for 5 min and cooled at 4 °C for 2 min. Afterward, 1.84 μL H_2_O, 1 μL NEB buffer 2, and 0.66 μL Klenow Fragment (3′→5′ exo-) were added for a total volume of 10 μL. The mixture was incubated at 37 °C for 40 min, 95 °C for 5 min, and 4 °C for 2 min. Finally, 0.66 μL Klenow Fragment (3′→5′ exo-) was added for a total volume of 10.66 μL, and the sample was incubated at 37 °C for 40 min.

#### 2.2.4. PCR Amplification

PCR amplification was conducted following the method outlined by Ofir et al. [25] with appropriate modifications using the QIAGEN Multiplex PCR Kit (QIAGEN, Hilden, Germany) on the products from Section 2.2.2 (RNA) and Section 2.2.3 (DNA). The SISPA primer (SISPA: GTTTCCCACTGGAGGATA) with 5′ phosphorylation was used. Then, 2.5 μL of the products from Section 2.2.2 or Section 2.2.3 was taken, and 12.5 μL 2X QIAGEN Multiplex PCR Master Mix, 2 μL 15 μM SISPA primers, and 8 μL H_2_O were added for a total volume of 25 μL. The following thermal cycling was performed: 95 °C for 15 min; 40 cycles (94 °C for 30 s; 56 °C for 30 s; 72 °C for 2 min); 72 °C for 15 min; 4 °C for 2 min. The amplified products had 5′ phosphorylation and a dA tail at the 3′ end, facilitating the subsequent attachment of the sequencing adapters.

#### 2.2.5. PCR Product Purification

RNA and DNA PCR amplification products (25 μL) were separately purified using the QIAquick PCR Purification Kit (QIAGEN, Hilden, Germany), with a final elution in 50 μL elution buffer. Subsequently, 45 μL of the nucleic acid from each purified product was combined with 18 μL AMPure XP Reagent (Beckman Coulter, Redwood, CA, USA) for further purification and concentration. The mixture was incubated at room temperature (18–23 °C) for 5 min, followed by two washes with 200 μL of 70% ethanol. Finally, the nucleic acids were eluted with 13 μL H_2_O. This two-step purification process utilizing the QIAquick PCR Purification Kit and AMPure XP Reagent is essential for removing large DNA polymer molecules generated after 40 cycles of PCR amplification, thereby reducing the occurrence of nanopore blockage during sequencing. The DNA concentration was determined by measuring 1 μL of the eluted nucleic acids using the Qubit 1X dsDNA HS Assay Kit (Thermo Fisher Scientific) on a Qubit 4 Fluorometer (Thermo Fisher Scientific).

#### 2.2.6. Preparation of Nanopore Sequencing Libraries

For library preparation, 100 ng of DNA was used with the Ligation Sequencing Kit (SQK-LSK110, Oxford Nanopore Technologies, Oxford, UK). Of the 100 ng of DNA, 50 ng originated from the purified product of RNA PCR amplification, and the other 50 ng originated from the purified product of DNA PCR amplification. If the Native Barcoding Kit 24 V14 (SQK-NBD114.24, Oxford Nanopore Technologies) was used, the DNA amount was increased to 150 ng, with 75 ng from RNA PCR amplification and 75 ng from DNA PCR amplification. The SQK-LSK110 kit allows for the sequencing of a single sample per run, while the SQK-NBD114.24 kit enables the barcoding of different samples with up to 24 unique barcodes on a flow cell for simultaneous sequencing.

Modifications were made to the kit protocols as necessary. Due to the use of 5′ phosphorylated primers and the presence of a dA tail generated after PCR amplification, the DNA repair and end-prep steps were omitted. Instead, the process proceeded directly to adapter ligation and clean-up (SQK-LSK110) or native barcode ligation (SQK-NBD114.24). The resulting DNA library was then eluted in 13 μL elution buffer. To determine the DNA library concentration, 1 μL of the DNA library was quantified using the Qubit 1X dsDNA HS Assay Kit on a Qubit 4 Fluorometer.

Since the PCR amplification products from different samples had an average length of approximately 600 bp, the library loading on the flow cell was optimized to maximize the utilization of effective nanopores without overloading the flow cell. For SQK-LSK110, 40–50 ng of the DNA library was taken for priming and loaded onto an R9.4.1 flow cell (FLO-MIN106D). For SQK-NBD114.24, 20–30 ng of the DNA library was used for priming and loaded onto an R10.4.1 flow cell (FLO-MIN114). The R10.4.1 flow cell is an improved version of the R9.4.1 flow cell, offering higher raw sequencing data accuracy (higher Q score), although both can meet the requirements of clinical diagnostics.

#### 2.2.7. Sequencing and Data Analysis

Sequencing was performed using the MinION sequencer connected to a computer equipped with NVIDIA GeForce RTX 3090 Graphics Cards(NVIDIA, Santa Clara, CA, USA). GPU-accelerated MinKNOW (Oxford Nanopore Technologies) was used for base calling in Super Accuracy mode. The Fastaq-pass files generated by basecalling were uploaded to the EP2ME (Oxford Nanopore Technologies) WIMP analysis program for real-time pathogen detection. If pathogenic organisms were identified, local analysis was initiated.

For the first step of the local analysis, NanoFilt (https://github.com/wdecoster/nanofilt accessed on 1 November 2023) with the parameters “--headcrop 24 --tailcrop 24” was used to remove the SISPA-N9 primer sequences. Subsequently, the minimap2 (https://github.com/lh3/minimap2 accessed on 1 November 2023) alignment tool was employed to align the NanoFilt-processed fastq files against the reference sequences of the pathogen, resulting in an SAM file. The SAM file was then converted to a BAM file using samtools (https://github.com/samtools/samtools accessed on 1 November 2023) and sorted. Ivar (https://github.com/andersen-lab/ivar accessed on 1 November 2023) with the parameters “-q 10 m 8” was used for the initial reference-based assembly in the next step, generating a fasta file. The generated sequences in this fasta file were examined. If a large portion of the sequence in the middle could not be assembled (typically poor assembly at the 5′ and 3′ ends), this suggested a significant difference between the reference sequence and the sample sequence. In such cases, it is advisable to reconsider the choice of the reference sequence. The sequences generated from the initial assembly were used to select a new reference sequence, and a BLAST search was performed against the NCBI database to identify the most closely related complete genome as the reference for the second round of reference-based assembly.

The relationship between different data volumes and assembly completeness was compared to determine the quantity of data required to assemble a relatively complete viral genome (>90%).

## 3. Results

### 3.1. Clinical Sample Detection Results

A total of nine clinical samples were collected and subjected to host gene removal treatment. After host gene removal, the SISPA method was employed to amplify DNA and RNA within the samples. Subsequently, metagenomic sequencing was conducted using the MinION platform. The results were compared with bacterial cultures and PCR tests to validate the method’s ability to detect various bacteria, DNA viruses, and RNA viruses in the samples, providing support for clinical diagnostics (Table 2). Other sequencing information can be found in the Appendix A.

PCR detected 15 viruses and 3 bacteria, and the results were consistent with the metagenomic sequencing results. Bacterial culture identified the same three bacteria as detected by metagenomic sequencing. In addition to these findings, metagenomic sequencing also detected some potential pathogens that could be associated with diseases. For example, a high level of RD114 retrovirus was detected in the serum of sample 1. No associations between RD114 retrovirus and disease have been reported, and further research is needed to determine whether a high viral load in the blood could lead to a decrease in the white blood cell count. Sample 3 contained *Clostridium difficile*, which is known to have strict culture requirements and is difficult to isolate. Sample 5 contained multiple pathogens; the metagenomic sequencing findings complemented the PCR and bacterial culture results and additionally detected *Mycoplasma hyorhinis*, *Glaesserella parasuis*, and PPV. Samples 6 and 7 contained PSaV and GETV, respectively. These two viruses are not commonly encountered clinical pathogens, and their presence was only discovered by metagenomic sequencing. These findings demonstrate that metagenomic sequencing can provide a more comprehensive assessment of the microbial pathogens present in a sample, especially clinically uncommon pathogens and those that are difficult to culture and isolate.

### 3.2. Host Gene Removal Efficiency

A comparative experiment was conducted on three clinical samples to assess the impact of host gene removal. The samples were compared to the group without host gene removal, i.e., no nuclease was added regarding pathogenic microorganism detection (Table 3).

Sample 5, which is lung tissue with a high content of host cells, showed a 10% reduction in host reads after nuclease digestion. Sample 9, which is ascites with a high protein content, showed a 9% reduction in host reads after nuclease digestion. Sample 8, a nasopharyngeal swab with low cell and protein content, showed a 50% reduction in host reads after nuclease digestion. After host depletion, all three samples showed a noticeable increase in the proportion of viral reads. The changes in the proportion of bacterial reads varied; however, all were detectable. The proportions of *Klebsiella pneumoniae*, *Mycoplasma hyorhinis*, and *Streptococcus suis* reads increased, while the proportions of *Pasteurella multocida* and *Glaesserella parasuis* reads decreased.

### 3.3. Sequencing Data Analysis

One challenge in the metagenomic sequencing of clinical samples is determining the appropriate volume of sequencing data. The fundamental clinical requirement is to identify the pathogens present in the sample, and the subsequent need is to obtain relatively complete genomes of these pathogens. According to the sequencing results of the pig and cat samples, it is evident that the real-time detection of multiple reads by EPI2ME WIMP can confirm the presence of a particular pathogen. However, the assembly of pathogen genomes differs between bacteria and viruses.

Bacterial genomes are relatively large, often several megabases in size, making it challenging to assemble complete genomes. In contrast, viral genomes are relatively small, typically a few kilobases, and the proportion of viral reads increases significantly after host gene removal, allowing for the assembly of relatively complete genomes and facilitating subsequent genetic evolution analyses. Target viral genome assembly was set at >90% completeness to determine when enough data had been generated and sequencing could be stopped, conserving flow cell usage, reducing sequencing time, and minimizing costs.

Because this sequencing method typically results in an N50 value of approximately 600 bp, multiplying 600 bp by the number of reads and then dividing by the size of the pathogen genome provides a rough estimation of the sequencing depth. A higher sequencing depth leads to greater completeness in viral genome assembly. For instance, if the FPV virus has a total length of 5.1 kb, when WIMP detects 5.1 k reads, the sequencing depth at that point is estimated to be 600X. Therefore, the number of detected viral reads and the size of the viral genome are the two main factors influencing the completeness of viral genome assembly. By examining the sequencing results for different viruses, we can identify when the number of viral reads is several times the size of the viral genome, allowing for the assembly of relatively complete viral sequences (>90%), thus indicating the optimal time to conclude sequencing (Table 4).

During the assembly of the genomes of the six viruses across the six samples, it was observed that when the number of reads was 0.025–0.2 times the full length of the reference genome, >90% of the genome could be assembled. At this point, the data generated from sequencing ranged from 3.3 to 98.3 MB, and the sequencing took only 5 min to 1 h. It is important to note that the data yield can be influenced by the condition of the MinION flow cell, with a new flow cell capable of producing 400–500 MB of data per hour. However, the data yield per hour gradually decreases with repeated use of the sequencing chip. Stopping the sequencing at this stage still provided most of the viral genome sequences and is a cost-effective choice. Continuing the sequencing process slowly increased the completeness of the sequences, mainly improving the completeness of the 5′ and 3′ ends. Few differences were observed when the sequences assembled from the >90% data volume were compared with those assembled from the total data volume; only a 2 bp difference in GETV was found. It is worth noting that the PCV2 genome is circular; thus, there are no challenges in amplifying the sequences at the 5′ and 3′ ends, and a lower data volume is required to obtain a 100% complete genome.

In the case of sample 1, which contained two different RD114 retroviruses, accurate sequences could not be assembled regardless of the data volume. The assembly showed continuous changes in the differing positions between the two viral strains as the volume of sequencing data changed; thus, this sample is not shown in Table 4. The PRRSV and PPV from sample 5 were also omitted because too few reads were obtained to make a comparison between different data volumes.

In summary, compared to bacterial culture and PCR testing, metagenomic sequencing demonstrates advantages in detecting difficult-to-culture bacteria and less common viruses. It offered the comprehensive detection of various pathogenic microorganisms present in the samples. After host gene removal, there was a notable increase in the viral genome content, while the proportion of bacterial genomes varied but remained detectable. The real-time monitoring of sequencing results allowed us to make informed decisions regarding when to stop sequencing, maximizing the utilization of sequencing chips and saving both time and costs.

## 4. Discussion

Nanopore sequencing, a third-generation sequencing technology, is characterized by its low equipment requirements and suitability for deployment in small laboratories. This sequencing strategy has been used in clinical sample testing [12,17,26]. In this study, we aimed to utilize optimized metagenomic nanopore sequencing and analysis methods to examine clinical samples from pigs and cats, assessing their performance in detection, host gene removal efficiency, and data analysis. This versatile metagenomic sequencing approach, applicable to various sample types, allowed for the simultaneous detection of bacteria, DNA viruses, and RNA viruses within 24 h. The results were consistent with the PCR testing and bacterial culture results, and several less common or difficult-to-culture pathogens were additionally detected, demonstrating the advantages of nanopore sequencing in clinical testing.

Clinical samples often have a substantial proportion of host genes, except for fecal samples. In this study, the method of grinding combined with nuclease digestion was employed, revealing a more effective enrichment of viral genes than bacterial genes. Although this method did not significantly increase the proportion of bacterial genomes, it did not impede bacterial detection and met the basic requirements for clinical testing. It is speculated that nuclease digestion, while removing a substantial amount of bacterial nucleic acids, may leave residual bacterial DNA or RNA detectable because bacterial genomes are relatively large, and this method simultaneously detects both DNA and RNA. However, further testing for bacterial virulence factors or resistance genes may not be achieved. For instance, in the case of sample 6, intestinal tissue, metagenomic sequencing detected *Escherichia coli* (commonly found in the gut microbiota); however, the data volume was insufficient for the detection of virulence genes, necessitating bacterial culture combined with PCR for identification.

Additionally, various factors influence the effectiveness of this method. For instance, samples with low cell and protein content (e.g., serum and nasal swabs, as seen in sample 8) tend to yield better host gene removal results because the HL-SAN nuclease can only digest free DNA and RNA, while intact nucleic acids protected within cells or by proteins remain undigested. Conversely, samples with a high cellular content, such as lung tissue (sample 5), and high protein content, such as cat ascites fluid (sample 9), showed lower proportions of host gene removal.

Furthermore, the freshness of the samples also plays a role in host gene removal, with samples collected freshly exhibiting better removal efficiency. In contrast, samples that have been stored under long-term freezing conditions or subjected to repeated freeze—thaw cycles may contain a substantial number of dead bacteria or viral nucleic acid that is not protected by the viral capsid. These organisms are susceptible to nuclease digestion during host gene removal. Therefore, in clinical testing, it is advisable to use fresh samples to achieve the best host removal and detection results [13,27].

This study introduced a method to estimate the required data volume for viral genome assembly, which sets it apart from other sequencing and analysis methods due to two factors. First, real-time detection can be performed using EPI2ME WIMP. Second, this sequencing method provides relatively stable N50 values when sequencing samples contain different viruses.

In this study, a reference-based assembly method was used for viral genome assembly. The advantage of reference-based assembly is that it requires relatively less data than de novo assembly. However, it has limitations when detecting newly emerging pathogens or those with significant genetic variations that are not present in the NCBI database. This study employed NanoFilt, minimap2, samtools, and ivar for reference-based assembly and found that ivar, originally developed for Illumina amplicon sequencing [28], was suitable for the reference-based assembly of random amplification nanopore sequencing data, providing satisfactory assembly results. This study used the ‘-m8′ parameter for ivar, which means that it considered sites with a sequencing depth greater than or equal to eight for reference-based assembly. This parameter was chosen because ‘-m8′ was deemed appropriate according to the available sample data. Increasing this value did not significantly improve assembly accuracy and reduced assembly completeness.

However, this study has some limitations. For instance, the number of clinical samples tested was limited, and future studies should expand the range and quantity of samples to further improve the sequencing and analysis methods. Additionally, the enrichment of bacterial genomes was not pronounced, and future researchers should consider combining this method with other host removal techniques, such as saponin and PMA, to enhance bacterial enrichment.

## 5. Conclusions

This study employed optimized metagenomic nanopore sequencing methods to examine various sample types from pigs and cats. These methods effectively detected bacteria and viruses in the samples and reduced the proportion of host genes. Combining cloud-based real-time and local analysis allowed rapid decision making regarding when to stop sequencing and enabled the assembly of relatively complete viral genomes. This method is suitable for application in small- to medium-sized veterinary clinical testing laboratories and provides valuable insights into the widespread use of metagenomic sequencing in veterinary clinical practice.

## Figures and Tables

**Figure 1 animals-13-03838-f001:**
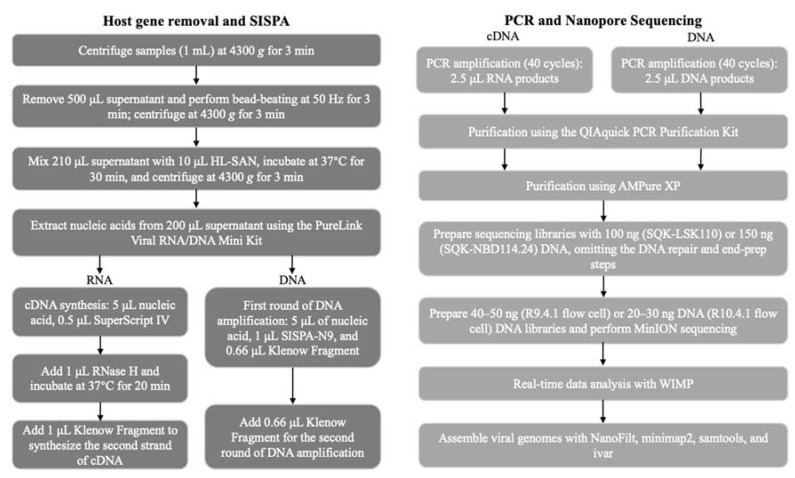
Schematic representation of the metagenomic pipeline. The turnaround time is approximately 16–24 h from sample collection to obtaining results.

**Table 1 animals-13-03838-t001:** Clinical sample information.

Sample No.	Sample Type	Clinical Symptoms	Detection Items
1	Cat Serum	Depression, hind limb weakness, decreased neutrophils, monocytes, eosinophils, and platelets	PCR ^1^: FPV, FHV-1, FCV, FCoV, FIV, FeLV Metagenomic Sequencing
2	Cat Nasopharyngeal Swab + Feces	Fever, increased nasal and ocular secretions, diarrhea	PCR ^1^: *Chlamydophila felis*, *Mycoplasma felis*, FPV, FCV, FHV-1 Metagenomic Sequencing
3	Pig Lung Tissue	Respiratory symptoms in sows, sudden death	PCR ^1^: CSFV, PRRSV, PCV2, PRV, ASFV Bacterial Culture ^2^ + 16S rRNA Sequencing ^3^ Metagenomic Sequencing
4	Cat Nasopharyngeal Swab	Respiratory symptoms	PCR ^1^: FCV, FHV-1, *Chlamydophila felis*, *Mycoplasma felis*, *Bordetella bronchiseptica* Metagenomic Sequencing
5	Pig Lung Tissue Pool (4 pigs)	Severe respiratory symptoms, fever, and high mortality in nursery pigs	PCR ^1^: CSFV, PRRSV, PRV, PCV2 Bacterial Culture ^2^ + 16S rRNA Sequencing ^3^ Metagenomic Sequencing
6	Pig Small Intestine Tissue	Diarrhea in piglets	PCR ^1^: PEDV, TGEV, PDCoV, PoRV Bacterial Culture ^2^ + 16S rRNA Sequencing ^3^ Metagenomic Sequencing
7	Pig Lung Tissue	Sudden death in piglets	PCR ^1^: CSFV, ASFV, PRRSV, PRV Bacterial Culture ^2^ + 16S rRNA Sequencing ^3^ Metagenomic Sequencing
8	Cat Nasopharyngeal Swab	Respiratory symptoms	PCR ^1^: FCV, FHV-1, *Chlamydophila felis*, *Mycoplasma felis*, *Bordetella bronchiseptica* Metagenomic Sequencing
9	Cat Ascites	Abdominal distention, ascites	PCR ^1^: FCoV Metagenomic Sequencing

^1^. PCR primer sequences are provided in the Appendix A. ^2^. Bacterial culture was performed using MacConkey agar, blood agar, and Columbia CNA blood agar plates. ^3^. Universal 16S rRNA primers were used for PCR, followed by Sanger sequencing. Primer sequences are provided in the Appendix A. FPV: feline parvovirus; FHV-1: feline herpesvirus 1; FCV: feline calicivirus; FCoV: feline coronavirus; FIV: feline immunodeficiency virus; FeLV: feline leukemia virus; CSFV: classical swine fever virus; PRRSV: porcine reproductive and respiratory syndrome virus; PCV2: porcine circovirus type 2; PRV: pseudorabies virus; ASFV: African swine fever virus; PEDV: porcine epidemic diarrhea virus; TGEV: transmissible gastroenteritis virus; PDCoV: porcine deltacoronavirus; PoRV: porcine rotavirus.

**Table 2 animals-13-03838-t002:** Detection results of clinical samples.

Sample No.	PCR Results	Bacterial Culture and Identification	Metagenomic Sequencing	Pathogenicity
1	FPV(−), FHV-1(−), FCV(−), FCoV(−), FIV(−), FeLV(−)	Not subjected to bacterial culture	RD114 retrovirus (80,572 reads; 45.3%)	The pathogenicity of the RD114 retrovirus is unknown; its relevance to leukopenia is uncertain
2	*Mycoplasma felis* (−), *Chlamydia felis* (−), FPV(+), FCV(−), FHV-1(−)	Not subjected to bacterial culture	FPV (22,210 reads; 12.9%)	FPV can cause fever and diarrhea in cats
3	CSFV(−), PRRSV(−), PCV2(−), PRV(−), ASFV(−)	*Pasteurella multocida*	*Pasteurella multocida* (11,520 reads; 1.8%) *Clostridium novyi* (1313 reads; 0.2%)	*Pasteurella multocida* can cause respiratory symptoms, and *Clostridium novyi* can cause sudden death in sows
4	FCV(−), FHV-1(−), *Mycoplasma felis* (+), *Chlamydia felis* (+), *Bordetella bronchiseptica*(−)	Not subjected to bacterial culture	*Mycoplasma felis* (14,404 reads; 18.7%) *Chlamydia felis* (5817 reads; 7.6%)	*Mycoplasma felis* and *Chlamydia felis* can cause respiratory symptoms in cats
5	CSFV(−), PRRSV(+), PRV(−), PCV2(+)	*Pasteurella multocida, Streptococcus suis*	PCV2 (47,944 reads; 10.9%) PRRSV (1934 reads; 0.4%) PPV (992 reads; 0.2%) *Mycoplasma hyorhinis* (21,555 reads; 4.9%) *Pasteurella multocida* (8857 reads; 2.0%) *Glaesserella parasuis* (3142 reads; 0.7%) *Streptococcus suis* (559 reads; 0.1%)	PCV2 can cause immunosuppression and slow growth. PRRSV can cause significant respiratory symptoms and death. PPV can cause reproductive disorders. *Mycoplasma hyorhinis*, *Pasteurella multocida*, *Glaesserella parasuis*, and *Streptococcus suis* can all cause respiratory symptoms.
6	PEDV(−), TGEV(−), PDCoV(−), PoRV(−)	*Escherichia coli*, but toxin gene test results were negative ^2^	PSaV (19,359 reads; 4.9%) *Escherichia coli* (5758 reads; 1.5%)	PSav can cause diarrhea in piglets
7	CSFV(−), ASFV(−), PRRSV(−), PRV(−)	No pathogenic bacteria grew	GETV (29,244 reads; 27.5%)	GETV can cause neurological symptoms and death in piglets
8	FCV(−), FHV-1(+), *Mycoplasma felis* (−), *Chlamydia felis* (−), *Bordetella bronchiseptica*(−)	Not subjected to bacterial culture	FHV-1 (165,200 reads; 47.2%)	FHV-1 can cause respiratory symptoms in cats
9	FCoV(+)	Not subjected to bacterial culture	FCoV (30,102 reads; 3.3%)	FCoV can cause ascites in cats

Pathogen reads represent the pathogen reads detected by WIMP. The percentage indicates the proportion of pathogen reads classified by WIMP. ^2^. Virulence factor testing of *Escherichia coli* was performed using PCR for heat-stable enterotoxin a (STa), heat-stable enterotoxin b (STb), heat-labile toxin (LT), shiga toxin type 2e (Stx2e), and cytotoxic necrotizing factor (CNF). The primer sequences can be found in the Appendix A. PPV: porcine parvovirus; PSaV: porcine sapovirus; GETV: Getah virus.

**Table 3 animals-13-03838-t003:** Comparison of sequencing results before and after host gene removal.

Sample No.	Host Removal	Host Reads (%)	PCV2 Reads (%)	PPV Reads (%)	PRRSV Reads (%)	FHV Reads (%)	FCoV Reads (%)	*Klebsiella pneumoniae* Reads (%)	*Mycoplasma hyorhinis* Reads (%)	*Streptococcus suis* Reads (%)	*Pasteurella multocida* Reads (%)	*Glaesserella parasuis* Reads (%)
5	No	77	1.10	0.010	0.00086	-	-	0.52	1.45	0.0095	6.24	1.73
	Yes	67	10.86	0.26	0.55	-	-	0.55	4.88	0.13	2.01	0.71
8	No	98	-	-	-	0.4	-	-	-	-	-	-
	Yes	48	-	-	-	47	-	-	-	-	-	-
9	No	82	-	-	-	-	1.6	-	-	-	-	-
	Yes	73	-	-	-	-	3.3	-	-	-	-	-

**Table 4 animals-13-03838-t004:** Viral genome assembly results.

Sample No.	Virus	Genome Type	Reference Strain NCBI ID	Reference Genome Size (bp)	Data Volume for >90% Completion ^1^	Completeness for >90% Completion ^2^	Viral Reads for >90% Completion /Reference Genome Size ^3^	Total Data Volume ^1^	Completeness for Total Data ^2^	Viral Reads /Reference Genome Size ^3^	Variant Bases ^4^ (bp)
2	FPV	Linear ssDNA	MT614366.1	5125	7.3 MB	92.9%	0.13X	240.5 MB	95.4%	4.3X	0
5	PCV2	Circular ssDNA	NC006232.1	1767	3.3 MB	100%	0.15X	589.4 MB	100%	27X	0
6	PSaV	Linear ssRNA	JX678943.1	7342	7.8 MB	91.9%	0.03X	630.8 MB	Near 100%	2.6X	0
7	GETV	Linear ssRNA	NC006558.1	11,597	6.5 MB	98.1%	0.13X	125.6 MB	99.6%	2.5X	2
8	FHV-1	Linear dsDNA	NC013590.2	135,797	98.3 MB	93.8%	0.2X	594.1 MB	98.8%	1.2X	0
9	FCoV	Linear ssRNA	MW030108.1	29,234	37.1 MB	95.3%	0.025X	1.6 GB	97.5%	1X	0

^1^. Data volume includes fastq.gz files with Q score ≥10. ^2^. Completeness is determined by comparing the assembled sequence to the reference sequence. ^3^. Viral reads for >90% completion/reference genome size. For example, for FPV using the data volume for >90% completion, the FPV reads number is 671, and the size of the FPV reference genome is 5125 bp; 671/5125 = 0.13X. ^4^. Variant bases represent the bases that differ between the sequences assembled with >90% data volume and the sequences assembled with the total data volume.

## Data Availability

The data presented in this study are available on request from the corresponding author. The data are not publicly available due to it will be used for subsequent epidemiological studies of the viruses.

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
