# Peer review of "Application of Host-Depleted Nanopore Metagenomic Sequencing in the Clinical Detection of Pathogens in Pigs and Cats"

_animals, 2023, doi:10.3390/ani13243838_

Round 1
Reviewer 1 Report
Comments and Suggestions for Authors
The authors present a “new/optimized” protocol for metagenomic sequencing of porcine and feline diagnostics samples. While the results are exiting, the manuscript lacks a complete representation of the field of (long-read) sequencing-based diagnostics. The protocol seems to be adapted from an existing protocol which requires a proper reference to the manuscript.Furthermore, the applied long-read sequencing is performed with incompatible flow cells which is not acceptable and will bias sequencing output, pore occupancy, and more importantly read quality, which then on its own impacts further downstream classification and genome assemblies. In addition, applied bioinformatics are not the generally accepted workflows in the field of read classifications and genome assemblies.
Abstract – The abstract is too extensive and should be shortened to be to the point.
Abstract – The authors talk about expensive equipment and complexity of sequencing but this is not reflected in the continuation of the text, what is the cost per sample and how is this less complex of a protocol?
Introduction – “Metagenomic sequencing has only recently been applied in animal clinical diagnostics [7−9].” This is not a complete reference list. There has been much work done in the laboratory of prof. Nauwynck at Ghent University on metagenomics-based diagnostics. They even have a spin-off company, PathoSense, which uses ONT for diagnostics. The next sentence should also be corrected as such.
Introduction – “One common issue in the metagenomic sequencing of clinical samples is that host genes comprise most of the data (e.g., up to 99.999% of the genomic information in sputum samples) [12, 13],” This is not correct for all sample types. Differences should be made between different sample types. Mucus and tissue samples have a much higher abundance as compared to lavages, serum, and fecal samples.
Introduction – “only one or two reads originate from the pathogen [14]” How to distinguish this from contamination or bioinformatics misclassification? What are generally applied cut-offs in this field?
Introduction – “Common host gene removal methods include mechanical grinding combined with nucleases [15, 16], saponin and nuclease treatment [12, 17−19], propidium monoazide (PMA) treatment [20, 21], and commercial reagent kits [17, 22].” Some important references in the field are missing here; e.g.,
https://www.nature.com/articles/s41598-018-28180-9
https://pubmed.ncbi.nlm.nih.gov/30128991/
Introduction – “Conventional next-generation sequencing platforms (e.g., Illumina and Ion Torrent) are expensive and often unaffordable for animal clinical diagnostic laboratories.” When making these statements more facts and figures should be included? ONT is not always cheaper than Illumina?
Introduction – “In this study, we employed optimized nanopore metagenomic sequencing and analysis methods to test various sample types from pigs and cats” Optimized from which protocol?
Table 1 – Please include the actual PRRSV type that was detected, same for the PCV2 and RoRV types as this is clinically important/relevant data. How was PRRSV (and others) distinguished from vaccination-based detection? For the cat ascites, which type of FCoV was identified?
Figure 1 – What does SISPA mean? The authors perform a centrifugation at 4,3000 xg on the samples. How sure are they no bacterial/fungal pathogens are missed by this centrifugation step? This seems to be taken from an existing protocol, so should be referred to as such. Using 40 PCR cycles seems a lot for metagenomic sequencing? How does this high number of cycles impact the actual (semi-quantitative) data output? Why do the authors perform both a purification on-column and with AMPure beads? This seems unnecessary to me? The authors should indicate why this is required? Why are the end-prep and DNQ-repair omitted? How many samples are pooled on a single MinION flow cell? I do not recommend the use of WIMP as it is known to be incomplete. There are various publications out there which show much better and more accurate read classification pipelines. This should be included in the manuscript as such. The use of NanoFilt is no longer used for genome assemblies, the use of Flye or canu is highly recommended for most accurate viral genome assemblies (others exist as well). Also, the use of read polishing (e.g., ONTs medaka) is of utmost importance to get to the most accurate nanopore-only genomes.
Section 2.2.1 – Why do the authors use linear acrylamide? The changes to the SISPA protocol (“appropriate modifications”) should be included properly.
Section 2.2.2– why do the authors use Klenow Fragments and not the more broadly applied strand switching oligo approach?
Section 2.2.3 – How were DNA concentrations determined and how was DNA/RNA QC done for the samples?
Section 2.2.5 – The use of AMPure beads at these concentrations will not generate in removal of HMW DNA? How was this verified experimentally? I highly recommend to perform a Tapestation analysis on the different fractions to confirm this actual removal and procedure as this seems not correct in my opinion.
Section 2.2.6 – The SQK-LSK110 and SQK-NBD114.24 kits are not compatible with the R9.4.1 kit. The LSK110 kit requires the R10.4 flow cell and the NBD114.24 kit requires the R10.4.1 flow cell. Also, loading ONT flow cells is usually expressed in molarity as it is highly dependent on fragment size. What’s the fragment sizes that are obtained from this protocol?
Comments on the Quality of English Language
Some sections and/or paragraphs require some English revision. E.g, some extensive sentences can be shortened to allow a better interpretation of the content.
Reviewer 2 Report
Comments and Suggestions for Authors
Nicely written paper with high practical diagnostic impact!
Only minor corrections:
Punctuation (comma, semicolon) under "Citation" should be reviewed.
Figure 1, right side: After cDNA and second strand synthesis plus RNaseH digestion there is no "RNA" any more that could be amplified.
2.2.1 line 13: do you mean a metal water bath?
3.3 line 10: host gene removal
Table4: legend: 671/5125 = 0,13x
Author Response
Thank you very much for taking the time to review this manuscript. Please find the detailed responses below.
Comments 1:Punctuation (comma, semicolon) under "Citation" should be reviewed.
Response 1: Thank you for pointing this out. I agree with this comment. Therefore, I have removed the extra commas.
Comments 2: Figure 1, right side: After cDNA and second strand synthesis plus RNaseH digestion there is no "RNA" any more that could be amplified.
Response 2: Agree. It can indeed lead to misunderstanding. I have changed 'RNA' to 'cDNA'
Comments 3: 2.2.1 line 13: do you mean a metal water bath?
Response 3: Thank you for pointing this out. I have changed to ‘Dry Bath incubator’.
Comments 4: 3.3 line 10: host gene removal. Table4: legend: 671/5125 = 0,13x
Response 4: Thank you very much. I have made the changes.
Reviewer 3 Report
Comments and Suggestions for Authors
Section 2.2.7, second paragraph: SISPA PCR often generates chimeric amplicons, so a simple tail and headcrop command in Nanofilt will not remove all SISPA primer sequences. Also, sequencing artefacts can be added to the ends of reads, and trimming 24 nt may not be enough to remove the primer sequence. I do not see a need to repeat the analysis because these sequencing artefacts are not significant when doing reference-based alignment of reads, but it is important to acknowledge in the discussion that removal of primer sequences using this method is not totally efficient.
Section 3.3 third paragraph: It must be acknowledged that a large proportion of reads are not viral reads, so this calculation method will provide only a rough estimation of the sequencing depth for the virus of interest.
Discussion: For viruses, dead is not the ideal term. It would be better to say that nucleases may digest viral nucleic acid that is not protected by the viral capsid.
It is important to acknowledge that the parameter "-m 8" in Ivar may generate a consensus sequence with spurious indels, since a depth of 8 reads is generally not enough for a reliable Nanopore consensus sequence. However, for general pathogen classification, lower depths can be used.
